# Pneumonia in Acute Ischemic Stroke Patients with Proximal Occlusions within the Anterior Circulation after Endovascular Therapy or Systemic Thrombolysis

**DOI:** 10.3390/jcm11030482

**Published:** 2022-01-18

**Authors:** Henning Muhl, Christian Roth, Andreas Schröter, Maria Politi, Maria Alexandrou, Janina Dahl, Susanne Gindorf, Panagiotis Papanagiotou, Andreas Kastrup

**Affiliations:** 1Department of Neurology, Klinikum Bremen-Mitte, St.-Jürgen-Street 1, 28177 Bremen, Germany; hanshenning.muhl@gesundheitnord.de (H.M.); andreas.schroeter@klinikum-bremen-ost.de (A.S.); janina.dahl@klinikum-bremen-mitte.de (J.D.); susanne.gindorf@gesundheitnord.de (S.G.); 2Department of Neuroradiology, Klinikum Bremen-Mitte, St.-Jürgen-Street 1, 28177 Bremen, Germany; Christian.roth@klinikum-bremen-mitte.de (C.R.); mariapoliti@hotmail.com (M.P.); maria.alexandrou@klinikum-bremen-mitte.de (M.A.); panagiotis.papanagiotou@klinikum-bremen-mitte.de (P.P.); 3Department of Neurology, University of Göttingen, Robert-Koch-Street 40, 37075 Göttingen, Germany

**Keywords:** stroke, thrombolysis, thrombectomy, outcome, stroke-associated pneumonia, risk factors

## Abstract

While endovascular treatment (ET) improves clinical outcomes in patients with proximal vessel occlusions compared to thrombolysis (IVT), the impact of ET on the frequency of stroke-associated pneumonia (SAP) is uncertain. We compared the rates of SAP in patients with large vessel occlusions in the anterior circulation after IVT or ET. We also determined risk factors for SAP, as well as the impact of SAP on early clinical outcomes. A total of 544 patients were treated with IVT, and 1061 patients received ET (with or without IVT). The rates of SAP did not differ significantly between ET (217/1061; 20%) and IVT (100/544; 18%) (*p* = 0.3). Overall, the occurrence of SAP was significantly associated with mortality and a poor clinical outcome. In the multivariable regression analysis, age, sex, the presence of dysphagia, early signs of ischemia on imaging and a history of stroke and mechanical ventilation were all significantly associated with the occurrence of SAP. In patients with large vessel occlusions, the introduction of ET did not result in lower rates of SAP compared with IVT. There is an ongoing need to reduce the rates of SAP in this patient population, for which the risk factors found here could become useful.

## 1. Introduction

After the publication of several landmark trials, endovascular treatment (ET) in addition to intravenous thrombolysis (IVT) has become the standard treatment for patients with proximal intracranial occlusions of the anterior circulation [1,2,3,4,5]. In these trials, ET was beneficial in all patient groups studied. A positive clinical effect of ET has also been observed in everyday clinical practice and, thus, outside of randomized trials [6]. In contrast to overwhelming clinical efficacy, the potential impact of ET on other stroke-associated complications has received little attention to date. In this context, the development of stroke-associated pneumonia (SAP) plays an important role. In fact, SAP is associated with multiple poor functional outcomes, including death (up to sixfold), neurological deterioration and a prolonged stay in hospital [7,8,9,10,11,12,13,14]. In contrast to the positive effect of ET on clinical outcome, the overall rates of SAP were high and comparable between IVT and ET in the MR CLEAN trial (11% after ET vs. 15% after IVT) [1]. Likewise, the rates of SAP did not differ significantly between ET or IVT in the ESCAPE trial (4.2% vs. 6%) [4] or the REVASCAT trial (14.6% vs. 8.7%) [3]. Therefore, the question arises as to whether ET reduces the frequency of SAP in addition to improving clinical outcomes compared with IVT. In addition, the relatively high rates of SAP in the aforementioned trials support the notion that there is an ongoing need to identify risk factors for this potentially devastating complication, especially in severely affected patients with large vessel occlusions.

The purpose of this study was to use our prospectively obtained large-volume stroke center database to compare the rates of SAP in patients with anterior circulation large vessel occlusions in two periods. During the first time period, all patients received systemic thrombolysis and, during the second time period, endovascular treatment with stent retrievers or aspiration devices was routinely used (with or without systemic thrombolysis). In the second step, we determined risk factors for SAP in this patient population.

## 2. Materials and Methods

### 2.1. Study Population

We identified patients with occlusions of the distal intracranial carotid artery and/or middle cerebral artery (M1 and M2 segment) who received treatment from January 2008 to December 2019. Patients with occlusions beyond the M2 segment were excluded. In addition, the current analysis excluded patients with posterior circulation strokes and bilateral strokes. Before November 2012 all patients had received systemic thrombolysis. The inclusion and exclusion criteria of thrombolytic therapy and the drug dose of our institution are mainly based on the NINDS study protocol. It can be treated up to 4.5 h after symptom onset, and there is no upper age limit for eligibility.

After November 2012, endovascular treatment with and without systemic thrombolysis (utilizing no upper age limit or specific imaging selection criteria) was routinely performed in patients who presented within 6 h (within 4.5 h in patients additionally treated with rt-PA) of symptom onset. At our institution ET is usually performed under conscious sedation, whilst general anesthesia is only used in very restless patients or those with limited vigilance.

In all patients, the following demographic data and stroke risk factors were gathered: age, gender, arterial hypertension, diabetes and atrial fibrillation. The time to thrombolysis (or to thrombectomy in patients without prior systemic thrombolysis) from stroke onset was likewise noted.

At our institution, all patients with acute cerebrovascular disorders are routinely screened for dysphagia within the first few hours after admission by a dysphagia-trained member of nursing staff. Patients who fail this initial screening remain “Nil by Mouth” until subsequently observed by a speech therapist within the first 24 h after admission. A standardized clinical evaluation based on the Munich Swallowing Score (MUCSS) for the assessment of the functional level of swallowing of saliva, fluids and food is used to quantify the severity of dysphagia [15]. In this study, the presence or absence of dysphagia was based on the results of speech therapists.

Our local stroke register has been approved by our local ethics committee (Ärztekammer Bremen) and is part of a nationwide mandatory quality assurance program for acute stroke treatment (Stroke Register of Northwestern Germany). The identity of individual patients is anonymous. Within this quality assurance program, no specific informed consent from patients or their relatives is required.

### 2.2. Imaging Techniques

Nonenhanced CT (NCT) and CT angiographic acquisitions before treatment were performed on a 4-row Multisection CT scanner (Siemens Volume Zoom, Siemens Medical Solutions, Forchheim, Germany).

NCT was performed with a patient in a head holder in the transverse plane. Incremental CT acquisitions of the brain were obtained by using the following parameters: 120 kVp, 250 mA, 2 s scan time and 5 mm section thickness.

CT angiography covered the region from the fifth vertebral body up to the vertex by using the following parameters: 120 kVp, 200 mAs, 4 × 1 mm collimation, 5.5 mm/rotation table feed and 0.5 s rotation time. CTA images were obtained after a single bolus contrast material (100 mL) injected intravenously at a flow rate of 4 mL/s using a power injector. In order to determine hemorrhagic, intracerebral complications as well as the final infarct sizes, CT or MR scans were obtained 1–3 days after treatment or immediately in the case of clinical worsening.

### 2.3. Image Analysis

All CT and MR images were analyzed jointly on a widescreen high-resolution monitor by 2 readers (a board-certified neuroradiologist with 21 years of neuroimaging review experience and a stroke neurologist with 19 years of neuroimaging review experience). With the exception of involvement, both readers were blinded to clinical and outcome data.

In order to detect early irreversible ischemia with a higher sensitivity than on plain CT images, the Alberta Stroke Programme Early CT Score was assessed on CTA source images (CTA-SI-ASPECTS) [16]. Follow-up CT or MR scans were used to determine the final ASPECTS as a marker of infarct extent, as well as the occurrence of symptomatic intracranial hemorrhages (SICH) by using the ECASS III definition.

### 2.4. End Points and Analyses

At our stroke center, the diagnosis of SAP is based on a combination of the following CDC criteria within 7 days of stroke onset [17]:Fever (<38 °C) with no other known cause;New onset of purulent sputum, change in character of sputum, increased respiratory secretions or increased suctioning requirements;New onset or worsening of cough, dsypnea or tachypnea with increased respiratory rates;Rales, crackles or bronchial breath sounds;Worsening gas exchange;Leukocytosis or leukopenia;Positive chest radiograph.

In a first step, we compared the rates of SAP after either IVT or ET. In a second step, we then determined clinical and radiological risk factors for SAP, regardless of treatment modality. In addition, we analyzed the impact of SAP on mRS at the time of hospital discharge.

### 2.5. Statistical Analysis

Continuous values were expressed as mean ± standard deviation (SD) or as median ± interquartile range (IQR). Nominal variables were expressed as counts and percentages. For comparisons of categorical data, two-tailed chi-square statistics with Yates correction and univariate Fisher’s exact test were used. The Fisher’s exact test was used when the predicted contingency table cell values were less than five. Analyses of continuous variables were performed with an unpaired Student’s *t*-test or, in cases of abnormally distributed data, with a Mann–Whitney U test.

A stepwise forward multiple-regression analysis was applied to determine the independent predictors for SAP. The following variables were considered: age; sex; initial NIHSS; treatment modality (i.v. thrombolysis alone, thrombectomy with thrombolysis and thrombectomy without thrombolysis); location of occlusion; presence of dysphagia; mechanical ventilation; history of a prior stroke; prestroke dependency (mRS ≥ 3); presence of hypertension; diabetes; hyperlipidemia; atrial fibrillation; baseline source image ASPECTS; and location in right or left hemisphere. Results are presented as odds ratios (ORs) with a 95% confidence interval (CI).

A value of *p* < 0.05 was considered to indicate a statistically significant difference. All statistical analyses were performed with SPSS (Version 22, SPSS Inc., Chicago, IL, USA).

## 3. Results

From January 2008 to December 2019, a total of 1605 patients with anterior circulation large vessel occlusions were treated (544 with systemic thrombolysis (IVT group) and 1061 patients with endovascular therapy (ET group)). Table 1 summarizes the baseline characteristics of both treatment groups. For the majority of risk factors, both treatment groups were well balanced. However, patients in the ET group were more severely affected and more often had early signs of ischemia than patients in the IVT group. These differences likely reflect higher rates of ICA/Carotid T occlusions and treatment times beyond 4.5 h or with unknown times of symptom onset in the ET group.

Significantly more patients in the ET group than in the IVT group had a favorable early clinical outcome (mRS 0–2)(32% vs. 26%; *p* < 0.05), as well as an excellent clinical outcome (mRS 0–1) (18% vs. 13%; *p* < 0.05). In addition, the final infarct sizes were significantly smaller after ET than after IVT (Table 2).

Overall, 317/1605 patients (19.7%) developed pneumonia. The rates of SAP did not differ significantly between ET (217/1061; 20%) and IVT (100/544; 18%) (*p* = 0.3). This result was also not affected when adjusting for potentially confounding variables between both treatment groups (OR_adjusted_, 1.1; 95% CI 0.8–1.5, *p* = 0.4).

Table 3 summarizes the baseline characteristics of patients with and without SAP.

Notably, patients with SAP were significantly older (78 years vs. 72 years, *p* < 0.001), more severely affected (initial NIHSS: 16 vs. 14, *p* < 0.001), more often had a dysphagia (89% vs. 58%, *p* < 0.001) and were significantly more often functionally impaired (mRS ≥ 3) before the current stroke event (22% vs. 12%, *p* < 0.001) than those without SAP, respectively. Atrial fibrillation and a prior stroke event were significantly more common in patients with SAP. Mechanical ventilation was also significantly associated with the rates of SAP.

Clinical and imaging outcome data for both groups are provided in Table 4. Patients with SAP were more likely to die within the hospital than patients without SAP (19% vs. 12%, *p* < 0.001). The rates of a poor outcome (mRS 5–6 at the time of discharge) were significantly higher in patients with SAP than without SAP (59% vs. 29%, *p* < 0.001). Vice versa, the rates of a good outcome (mRS 0–2 at the time of discharge) were significantly lower in patients with SAP compared to those without SAP (5% vs. 36%, *p* < 0.001). SAP was also associated with larger infarcts and a longer stay in hospitals.

In multivariate regression analysis, age, sex, the presence of dysphagia, early signs of ischemia on imaging (i.e., the SI-ASPECTS) and a history of stroke and mechanical ventilation were all significantly associated with the occurrence of SAP (Table 5).

## 4. Discussion

Based on a large dataset, the frequency of SAP in patients with large-vessel intracranial occlusions within the anterior circulation was similar between systemic thrombolysis alone and endovascular therapy with or without thrombolysis. Regardless of the treatment modality, we identified several risk factors for SAP in this patient population, including age, sex (male), the presence of dysphagia, early signs of ischemia on imaging and a history of stroke and mechanical ventilation.

In our patient population approximately 20% of all patients developed SAP after either IVT or ET. This high rate of SAP is within the range reported in previous randomized trials dealing with comparable patient populations [1,2]. In the MR CLEAN trial, the overall rate of SAP in patients with large vessel occlusions in the anterior cerebral circulation was 13% [1]. In the SWIFT PRIME trial, nearly 20% of the patients developed SAP after ET [2]. In contrast, the rates of SAP after either IVT or ET were lower in the ESCAPE trial (approximately 5%), likely reflecting a highly selected patient population with less comorbid conditions [4].

In all aforementioned trials, the rates of SAP in patients with proximal vessel occlusions were comparable after ET or IVT, despite the fact that ET improved clinical outcomes. In good agreement with this finding, the rates of SAP were nearly identical after either ET or IVT in this patient population in our case series. As in the randomized trials, ET also improved clinical and radiological outcomes in our series, compared with IVT. Taken together, these results clearly show that improved clinical outcome after ET does not automatically translate into reduced rates of SAP.

In broader stroke-patient populations, SAP has repeatedly been associated with poor clinical outcomes, including death [8,9,14]. Likewise, we observed a close relation between SAP and poor outcomes in patients with large vessel occlusions after either IVT or ET, supporting the notion that pneumonia is an important modifiable risk factor in this patient population. Future studies should, thus, address the question as to whether the use of prophylactic antibiotics could further improve clinical outcomes after ET in patients at high risk of SAP. Irrespective of treatment modality, several factors, including advanced age, sex, dysphagia, mechanical ventilation and early signs of ischemia were significantly associated with SAP in our study. Comparable factors have been identified in previous studies dealing with broader stroke-patient populations and have also been incorporated into risk scores for SAP [18,19]. The increased risk of SAP in older patients after either ET or IVT likely reflects higher instances of comorbid conditions, as well as impairments of swallowing and cough reflexes [20]. As in previous studies, we found an inverse association between female sex and SAP [21]. Although the reasons for the lower rates of SAP in females remain unclear, it could be speculated that women have a healthier lifestyle than men with lower rates of smoking and less alcohol consumption. In addition, female hormones could also contribute to this finding. In fact, female hormones have been shown to inhibit inflammatory reactions to bacterial antigens und could, thus, reduce the severity of respiratory infections [22]. Female hormones have also been associated with a more vigorous immune response, at least in animal models [22]. On the other hand, the vast majority of our female patients were in their menopause, suggesting that further, currently unknown sex-specific mechanisms must also contribute to the lower risk of SAP in females.

As expected, and in good agreement with several previous reports, mechanical ventilation was also significantly associated with the risk of pneumonia in our cohort in both treatment groups [23,24]. Against the background of this increased risk, there is an ongoing discussion as to whether ET should be performed under conscious sedation or general anesthesia [24]. However, our data should be interpreted with caution, since mechanical ventilation was only applied in severely affected patients, thus likely reflecting a selection bias.

In the past few years, several clinical risk scores for predicting SAP in broader stroke populations have been developed [25]. The majority of these predictive risk models were based on routinely available variables and commonly included age, atrial fibrillation, dysphagia, stroke severity as determined by the NIHSS and sex. It is noteworthy that NIHSS and atrial fibrillation were not significantly associated with SAP in the multivariate analyses in our study. This finding is likely attributable to the fact that our stroke patients with large vessel occlusions were more severely affected and more often had a cardio-embolic stroke due to atrial fibrillation than the patients entered in the aforementioned risk models.

The major strengths of our study include the large dataset and the use of uniform treatment algorithms. However, the present report also has limitations. The data were obtained retrospectively in a single academic center, and the impact of ET or IVT on SAP was determined in a nonrandomized fashion. Hence, the study method holds all drawbacks of observational design and limits the generalizability of our results. Several important risk factors for SAP were not considered in our analyses including details on nasogastric tube interventions, laboratory markers and the use of certain medications, which have been linked to the presence or absence of SAP [26,27,28]. Furthermore, dysphagia was only dichotomized as either absent or present, thus using a rather broad classification scheme of this important risk factor for SAP. In addition, we did not consider the time of onset and timecourse of pneumonia, which play a crucial role in the interpretation of stroke and pneumonia with respect to causality, especially in mechanically ventilated patients and those with signs of pneumonia already at the time of admission. Although we studied a rather large patient population and found comparable SAP rates after ET or IVT, we might have missed a significant difference between both groups. Using our data, however, more than 9000 patients in each treatment group would be needed to detect a potential significant difference, making a Type II error unlikely. Finally, we used mRS at the time of discharge to assess clinical outcome instead of the widely accepted 3 months.

## 5. Conclusions

In patients with acute large vessel occlusions, the overall frequency of SAP is high in everyday clinical practice and comparable between ET and IVT. Regardless of treatment modality, SAP is significantly associated with mortality and an overall poor clinical outcome. Thus, there is an ongoing need to reduce the rates of SAP in this patient population, for which the risk factors found in this study could become useful.

## Figures and Tables

**Table 1 jcm-11-00482-t001:** Baseline characteristics of acute ischemic stroke patients with large vessel occlusions in the anterior circulation after either thrombolysis or endovascular therapy.

	IVT	ET	*p*-Value
	(*n* = 544)	(*n* = 1061)	
Age	74 ± 12	74 ± 13	0.2
Mean (yrs)			
Female	294 (54)	569 (54)	0.9
Hypertension	451 (83)	858 (81)	0.3
Atrial fibrillation	270 (50)	488 (46)	0.2
Diabetes	112 (21)	223 (21)	0.8
Prior stroke	83 (15)	191 (18)	0.2
Prestroke dependency	89 (16)	142 (13)	0.1
NIHSS *	13 (10–17)	15 (11–18)	<0.01
(median, IQR)			
Dysphagia	353 (65)	673 (63)	0.5
Imaging			
Side of Occlusion			
Right hemisphere	255 (47)	514 (48)	0.6
Location of Occlusion **			
ICA	78 (14)	172 (16)	0.3
Carotid-T	17 (3)	153 (14)	<0.01
M1 MCA	301 (55)	555 (53)	0.2
M2 MCA	148 (27)	181 (17)	<0.01
Baseline SI-ASPECTS ***			
(median, IQR)	9 (7–10)	8 (7–10)	<0.05
Mechanical ventilation	38 (7)	99 (9)	0.1

* NIHSS = National Institutes of Health Stroke Scale; ** most proximal occlusion location; *** SI-ASPECTS = Source image Alberta Stroke Program Early CT Score (only for anterior circulation strokes); ICA: indicates internal carotid artery; MCA: indicates middle cerebral artery.

**Table 2 jcm-11-00482-t002:** Clinical and radiological outcomes according to treatment.

	Endovascular Therapy	Thrombolysis Alone	*p* Value
	*n* = 1061	*n* = 544	
Clinical outcomes			
mRS * 0–2	336 (32)	142 (26)	<0.05
mRS 0–1	190 (18)	73 (13)	*p* < 0.05
mRS 5–6	346 (33)	222 (41)	*p* < 0.05
Death	139 (13)	73 (13)	*p* = 0.9
Radiological outcome			
Infarct size **	7 (5–8)	6 (3–8)	*p* < 0.01
(Median, IQR)			

* mRS = modified Rankin Scale; ** using the follow-up ASPECTS = Alberta Stroke Program Early CT Score.

**Table 3 jcm-11-00482-t003:** Baseline characteristics of patients with and without stroke-associated pneumonia.

	Non-SAP	SAP	*p*-Value
Age Mean (yrs)	72 ± 13	78 ± 10	<0.01
Female	705 (55)	158 (50)	0.1
Hypertension	1023 (79)	266 (84)	0.07
Atrial fibrillation	591 (46)	167 (53)	<0.05
Diabetes	257 (20)	78 (25)	0.07
Prior stroke	200 (15)	74 (23)	<0.001
Prestroke dependency	161 (12)	70 (22)	<0.001
NIHSS * (median, IQR)	14 (10–17)	16 (13–19)	<0.001
Dysphagia	744 (58)	282 (89)	<0.001
Imaging			
Side of Occlusion			
Right hemisphere	620 (48)	150 (48)	1
Location of Occlusion **			
ICA	192 (15)	59 (19)	0.1
Carotid–T	134 (10)	35 (11)	0.7
M1 MCA	698 (54)	158 (50)	0.5
M2 MCA	264 (20)	65 (20)	1
Baseline SI–ASPECTS *** (median, IQR)	9 (7–10)	8 (6–10)	<0.001
SICH	46 (4)	13 (4)	0.6
Treatment			
i.v. Alteplase alone	444 (34)	100 (31)	0.7
Endovascular therapy			
with thrombolysis	476 (40)	126 (40)	1
Endovascular therapy			
without thrombolysis	368 (28)	91 (29)	1
Mechanical ventilation	89 (7)	48 (15)	<0.001

* NIHSS = National Institutes of Health Stroke Scale; ** most proximal occlusion location; *** SI-ASPECTS = Source image Alberta Stroke Program Early CT Score (only for anterior circulation strokes); ICA indicates internal carotid artery; MCA indicates middle cerebral artery.

**Table 4 jcm-11-00482-t004:** Clinical and imaging outcomes in patients with and without SAP at the time of discharge.

	Non-SAP (*n* = 1288)	SAP (*n* = 317)	*p*-Value
mRS at discharge (mean ± SD)	3.3 ± 1.8	4.6 ± 1.1	<0.001
mRS 0–2 (*n*; %)	461 (36)	17 (5)	<0.001
mRS 5–6 (*n*; %)	380 (29)	188 (59)	<0.001
Death (*n*; %)	151 (12)	61 (19)	<0.001
Infarct size * (median, IQR)	7 (5–8)	5 (3–7)	<0.001
Length of hospital stay (days) (median; IQR)	6 (4–9)	8 (6–12)	<0.001

mRS modified Rankin scale; ***** using the follow-up Alberta Stroke Program Early CT Score.

**Table 5 jcm-11-00482-t005:** Multivariable odds ratios and 95% confidence intervals of SAP in patients with large vessel occlusions in the anterior circulation.

	Odds Ratio	CI Lower	CI Higher	*p* Value
Age *	1.03	1.01	1.04	<0.001
Female	0.6	0.5	0.9	<0.01
Initial NIHSS **	1.02	0.99	1.1	0.09
Dsyphagia	4.5	3.1	6.8	<0.001
Prior stroke	1.7	1.2	2.5	<0.01
Baseline SI-ASPECTS ***	0.9	0.8	0.9	<0.05
Mechanical ventilation	2	1.3	3.2	<0.01

CI, confidence interval; NIHSS, National Institutes of Health Stroke Scale; * per 1 year increase; ** per 1 point increase; *** per 1 point increase.

## Data Availability

The data that support the findings of this study are available from the corresponding author upon reasonable request.

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
