# Peer review of "Pneumonia in Acute Ischemic Stroke Patients with Proximal Occlusions within the Anterior Circulation after Endovascular Therapy or Systemic Thrombolysis"

_jcm, 2022, doi:10.3390/jcm11030482_

Round 1

Reviewer 1 Report

The paper by Muhl et al. deals with the rate of pneumonia as complication of stroke in patients experiencing large vessel occlusion treated either with systemic thrombolysis (ST) or local endovascular thrombectomy (ET). The main result is that, despite the treatment, the rate of stroke associated pneumonia (SAP) is similar.

The study design is a single-center retrospective approach that is a limitation, as the authors correctly acknowledge in the discussion. The straight is the large number of patients included.

By reading the text I find some major limitations that have been overlooked in the current presentation.

The first one is that authors speculate several times that it is interesting that SAP is similar despite the treatment used, but this datum is mostly related to the literature. In the present paper, we don’t know if patients treated with ST have in the present population a worse outcome compared to those treated with ET. This datum should be added or discussed as a limitation.

Secondly authors found that NIHSS is not a predictor of SAP in their sample. This statement should be discussed better highlighting that this is only limited to a portions of the stroke patients and should not be extrapolate to other stroke populations. In fact, recent papers found that NIHSS was the best predictor of pneumonia (Westendorp WF Eur Stroke J. 2018 Jun;3(2):136-144; Jannini TB Neurol Sci. 2021 Jul 16). In both these papers NIHSS was found to be directly responsible for developing SAP. The fact that NOHSS is not directly related to SAP is probably due to the high level of NIHSS scores that both groups have.

Moreover, NIHSS is one of the major causative factor for dysphagia which was correctly identify by the authors as a predictor of SAP. In fact, a NIHSS equal to 4.5 was found to be the best cut-off between dysphagic and non-dysphagic patients by Henke C Cerebrovasc Dis (2017) 44(5–6):285–90, while a NIHSS score of 11.5 to be associated to persistent dysphagia and thus patients at higher risk to develop SAP (Toscano M Eur Neurol (2015) 74(3–4):171–7). This reasoning would reinforce the datum of dysphagia as a risk factor for SAP. In fact, in the present paper authors do not clarify as dysphagia was assessed neither the training in dysphagia detecting of the personnel involved (that I suppose to be ward regular nurses). This limitation should be discussed in the discussion since we currently have a large number of validate scales to measure dysphagia in stroke patients and some of them should be routinely used to assess dysphagia, as the TOR-BSST (Martino R Stroke. 2009;40(2):555-61), GLOBE-3S (Toscano M Eur J Neurol. 2019;26(4):596-602), BSTD (Immovilli P J Stroke Cerb Dis 2021;30:105470), GUSS (Tralp M Stroke 2007;38:2948-2952). The existence of these scales should be mentioned and discussed.

Reviewer 2 Report

I am grateful for the opportunity to review this manuscript. The authors compared the frequency of pneumonia in two subsets of stroke patients. Their results show no difference in the frequency of pneumonia.

There are some issues that, from my perspective, need either correction or clarification if this manuscript is to be published:

  1. The title suggests that the authors investigated patients with stroke-related pneumonia. Nevertheless, they studied patients with ischemic stroke (IS) exclusively. Thus the population under study is not correctly identifiable in the title. 
  2. The study's objective is established in terms of incidence; nevertheless, in the abstract and several other manuscript locations, the authors declare that they obtained a rate. In different sections, they report the frequency. Furthermore, the authors seek to identify risk factors for the development of pneumonia. It is worth noting that it is possible to find in the published literature data regarding the principal risk factors for developing pneumonia after IS. Some of the identified risk factors are also described in this manuscript, but surprisingly, some were not. Examples of these are: atrial fibrillation (odds ratio (OR) = 2.884, 95% confidence intervals (CI) = 1.316-6.322)- Yuan et al. 2021) and astonishingly being subject to an invasive procedure (OR = 12.838, 95%CI = 6.296-26.178 - same reference). I think that the authors should address these significant differences.
  3. Given that the risk factors for pneumonia are already known, I suggest that identifying risk factors for pneumonia should not be an objective of this study but a source of variables to account for in the comparisons between both groups (IVT vs. ET). By comparing paired cases matched for the presence of the known risk factors for pneumonia and matched for IS severity and premorbid disability, the authors increase their chance of finding a meaningful difference between treatment groups. In its current form, the manuscript text and data do not support the conclusion that the overall incidence of pneumonia is comparable between ET and IVT.
  4. According to the study's primary objective, the analysis, results, and tables should reflect comparisons among the patients by treatment group, not between the presence of pneumonia.

Reviewer 3 Report

I read with interest the paper " Stroke-associated pneumonia in patients with proximal occlusions within the anterior circulation after endovascular therapy 3 or systemic thrombolysis”.

Please find my comments and suggestions below:

  1. Page 1, line 23, I think that multivariable models were perfumed not multivariate.
  2. Line 138, please specify, which regression was used.
  3. “(P in .05, P out .1)” what if P value was between 0.05 and 0.1?
  4. Table 2, please check the value for mRS at discharge.

Round 2

Reviewer 2 Report

The manuscript has substantially improved since the last version. Nevertheless, I strongly encourage the authors to clarify and align the epidemiological terms used throughout the text.

In this regard, incidence refers to the occurrence of new cases of disease or injury in a population over a specified period of time. Moreover, two types of incidence are commonly used — incidence proportion and incidence rate, where incidence rate (also known as person-time rate) is a measure of incidence that incorporates time directly into the denominator. Whereas frequency is a broader term that encompasses common frequency measures such as ratios, proportions, and also rates. (Last JM. A dictionary of epidemiology, 4th ed. New York: Oxford U. Press; 2001.)

Author Response

According to the suggestion of this reviewer we have aligned the epidemiological terms throughout our manuscript and now use "frequency" and "rates" instead of "incidence".